# Weighted Ensemble Object Detection with Optimized Coefficients for Remote Sensing Images

**Atakan Körez** [1],*[ID], **Necaattin Barışçı** [1][ID], **Aydın Çetin** [1][ID] and **Uçman Ergün** [2]

[1] Department of Computer Engineering, Faculty of Technology, Gazi University, Ankara 06560, Turkey; nbarisci@gazi.edu.tr (N.B.); acetin@gazi.edu.tr (A.Ç.)

[2] Biomedical Engineering Department, Afyon Kocatepe University, Afyon 03300, Turkey; uergun@aku.edu.tr

\* Correspondence: atakan.korez@gazi.edu.tr

**Abstract:** The detection of objects in very high-resolution (VHR) remote sensing images has become increasingly popular with the enhancement of remote sensing technologies. High-resolution images from aircrafts or satellites contain highly detailed and mixed backgrounds that decrease the success of object detection in remote sensing images. In this study, a model that performs weighted ensemble object detection using optimized coefficients is proposed. This model uses the outputs of three different object detection models trained on the same dataset. The model's structure takes two or more object detection methods as its input and provides an output with an optimized coefficient-weighted ensemble. The Northwestern Polytechnical University Very High Resolution 10 (NWPU-VHR10) and Remote Sensing Object Detection (RSOD) datasets were used to measure the object detection success of the proposed model. Our experiments reveal that the proposed model improved the Mean Average Precision (*m*AP) performance by 0.78%–16.5% compared to stand-alone models and presents better mean average precision than other state-of-the-art methods (3.55% higher on the NWPU-VHR-10 dataset and 1.49% higher when using the RSOD dataset).

**Keywords:** aerial object detection; deep learning; remote sensing; ensemble object detection

## 1. Introduction

At present, with the enhancement of remote sensing technologies, remote sensing image object detection has gained popularity thanks to its successful civil and military applications (e.g., urban monitoring, traffic monitoring, agricultural applications, and landscape planning). However, various difficulties remain in the field of remote sensing object detection, such as complicated and diverse views, the high costs of manual annotation, and differences in the instant detection of large scene images. Besides that, a typical feature of high-resolution images from planes or satellites is a highly detailed and mixed background. Object detection methods that use deep learning methods (which are increasing in popularity) have reached a state-of-art level [1].

Many studies on the detection of objects in remote sensing images are available in this field of study. For instance, Peicheng et al. [2] proposed a novel and effective approach to train a Rotation-Invariant Convolutional Neural Network (RICNN) model for advancing the performance of object detection, and Wang et al. [3] used a skip-connected encoder–decoder model to extract multiscale features from a full-size image. Wu et al. [4] detected remote sensing objects using Fourier-based rotation-invariant feature boosting (FRIFB). Dong et al. [5] used the Sigmoid Nonmaximum Suppression (Sig-NMS) method instead of the traditional nonmaximum suppression (NMS) to reduce the rate of undetectability for small targets. Gong et al. [6] introduced an object detection model that enriches feature representation and adopts the basic context information between objects. Cheng et al. [7] proposed a multiclass object detection feedback network (MODFN) using a top-down feedback mechanism based on a

traditional feedforward network. Wang et al. [8] achieved object detection using a unified framework that can gather contextual information at multiple scales along with feature maps at the same scale. Zhang et al. [9] proposed a Double-Net model with multiple CNN channels, where each channel corresponds to a certain direction of rotation, and all CNNs share the same weights. Cheng et al. [10] developed a rotation-invariant framework based on the Collection of Part Detectors (COPD) for multiclass object detection. Li et al. [11] proposed a dual-channel feature fusion network capable of learning regional and contextual properties along two independent paths. Wang et al. [12] proposed a new region proposal network that is anchor-free and sliding-window-free to develop a two-stage object detection network. Körez et al. [13] improved the Faster R-CNN algorithm via the deformable convolution and the weight standardization techniques for remote sensing object detection using low-capacity Graphic Processing Units (GPUs). Qui et al. [14] proposed a novel end-to-end Adaptive Aspect Ratio Multiscale Network ($A^2$RMNet), which is a multiscale feature gate fusion network used to adaptively integrate the multiscale features of objects. Xu et al. [15] proposed a CNN structure called deformable ConvNet, which adds offsets to the convolution layers to address geometric modeling in object recognition. Zhou et al. [16] proposed an encoder–encoder architecture called a rotated feature network (RFN), which produces rotation-sensitive feature maps (RS) for regression and rotation-invariant feature maps (RI) for classification. For a comprehensive and recent survey, see [17].

In recent years, studies using ensemble models have been prominent for object detection in remote sensing images [18–23]. In these studies, multiple multilayer perceptions, conditional random fields, or CNN ensembles were used to create structurally complete models. Then, the models were trained, and object detection was performed.

All existing models are based on specific structural conditions. Therefore, it is difficult to make changes to the models. This led researchers to focus on training ensemble models. However, training an ensemble model is not practical. However, in all of these studies, training was carried out on such ensemble models. A training-independent ensemble operation would be more practical and effective. In this study, we propose a flexible and training-independent weighted ensemble object model that uses three different object detection models (Single Shot Multibox Detector (SSD) [24], RetinaNet [25], and Improved Faster Regional Convolutional Neural Network (Faster R-CNN) [26]) that are independently trained on the same dataset. The SSD, RetinaNet, and Faster R-CNN models have achieved state-of-art success in object detection and are pioneers in the field. We chose the SSD, RetinaNet, and Improved Faster R-CNN models because they can optimize Faster R-CNN and previously achieved very successful results on the Northwestern Polytechnical University Very High Resolution 10 (NWPU-VHR10) and Remote Sensing Object Detection (RSOD) datasets. Our contributions are as follows:

- A weighted ensemble object detection model using three different object detection models trained on the same dataset is proposed;
- The proposed model is independent of the object detection models it uses as input. This means that two or more different types of object detection models can be used in the proposed model. Since our model has a weighted ensemble structure, the number of different structures of the models used as input will not affect the overall structure of the model;
- Our study is the first to perform object detection of remote sensing images using multiple object detection models as coefficient-weighted ensembles with optimized coefficients that are independently trained on the same dataset.

In the second section of this study, the materials and methods we used are described. In the third section, the proposed optimized coefficient-weighted ensemble model is outlined. The fourth section focuses on the test environment, evaluation criteria, and explanation of the datasets (NWPU-VHR10 and RSOD) that we used to measure the performance of our proposed model, as well as experiments using different coefficients (including optimal coefficients) and a comparison of our model with other

studies using the same datasets. The conclusions of the study are summarized in Section 5, and Section 6 discusses future work.

## 2. Methods

In this section, the structures and concepts used by the remote sensing object detection model introduced in our study are explained.

### 2.1. Single Shot Multibox Detector (SSD)

The SSD divides the output area of the bounding boxes into a series of default boxes with different aspect ratios and scales for each feature map location [24]. While making predictions, the algorithm generates points for the presence of each category of objects in the default box and adjusts the sizes of the boxes to better match the shape of the object. In addition, predictions from multiple attribute maps with different resolutions are combined so that objects of different sizes can be handled easily (Figure 1).

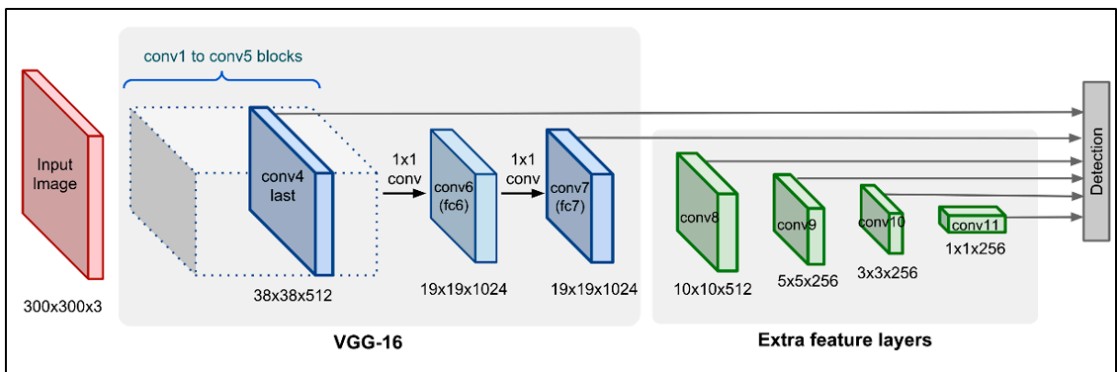

**Figure 1.** Block diagram of the Single Shot Multibox Detector (SSD).

### 2.2. RetinaNet

RetinaNet is a single unified network consisting of a backbone network and two task-specific subnets. To increase the object detection success, RetinaNet uses Focal Loss [25], which is a new type of loss that reduces the relative loss of well-classified samples and pays more attention to difficult-to-detect and misclassified samples. In RetinaNet, the backbone is responsible for calculating a convolutional property map over the entire input image. The first subnet classifies the backbone output, and the second subnet performs the convolution, thereby limiting box regression (Figure 2).

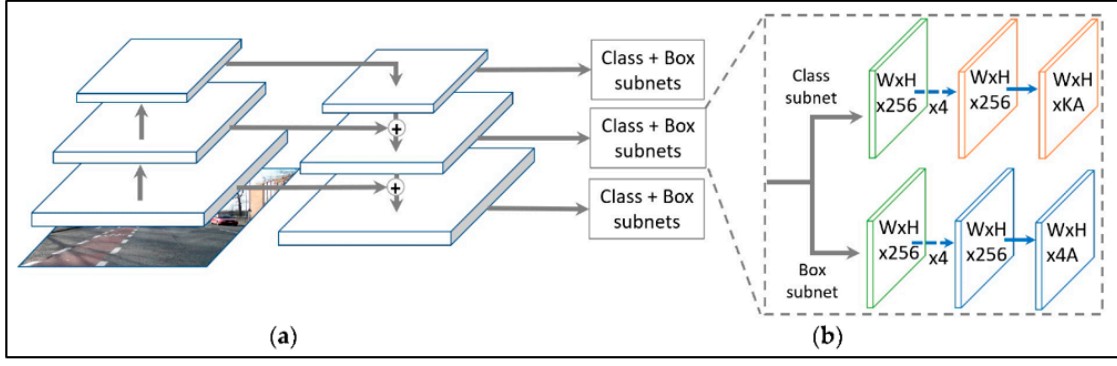

**Figure 2.** Architecture of RetinaNet: (**a**) first subnet and (**b**) second subnet.

*2.3. Improved Faster R-CNN*

The present model is based on the latest Faster R-CNN (Figure 3) [26], a state-of-the-art object detection system. The deformable convolution [27] technique is used to overcome the weaknesses of the regular convolution structure used in the Faster R-CNN model for detecting small and mixed objects in remote sensing images [13]. With the Feature Pyramid Network (FPN) [28] technique, the high-resolution features in the shallow layers of the remote sensing images are transferred to the network. The Weight Standardization (WS) [29] technique, which reduces the batch size to provide deep learning training without performance problems on low power / single GPU systems, such as those using a single GPU, was also added to the model.

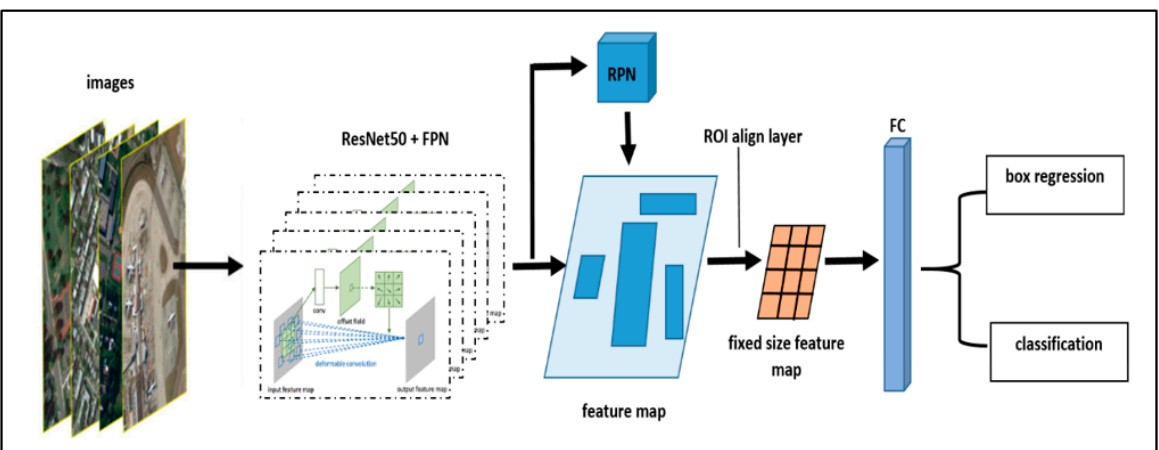

**Figure 3.** Network structure of the Improved Faster R-CNN.

*2.4. General Concepts of Object Detection*

In this subsection, the concepts we use within the scope of object detection are briefly defined. Those are;

- Bounding Boxes: These are the rectangular boxes drawn around the objects to make the detected objects more prominent after the object detection is complete;
- Common Box: This is the box used to express the bounding boxes from three different object detection models, which are used as the input for the Weighted Ensemble Block; a single bounding box is called common box;
- Confidence Score: This is the probability that an anchor box contains an object and is usually predicted by a classifier. The confidence score contains cut-off predictions [26–28].

**3. Proposed Model**

In this section, we describe the ensemble algorithm that we used to combine the included object detection models, the optimization of the coefficients, and the outputs of these object detection models. Figure 4 shows a block diagram of the object detection model that we propose in our study.

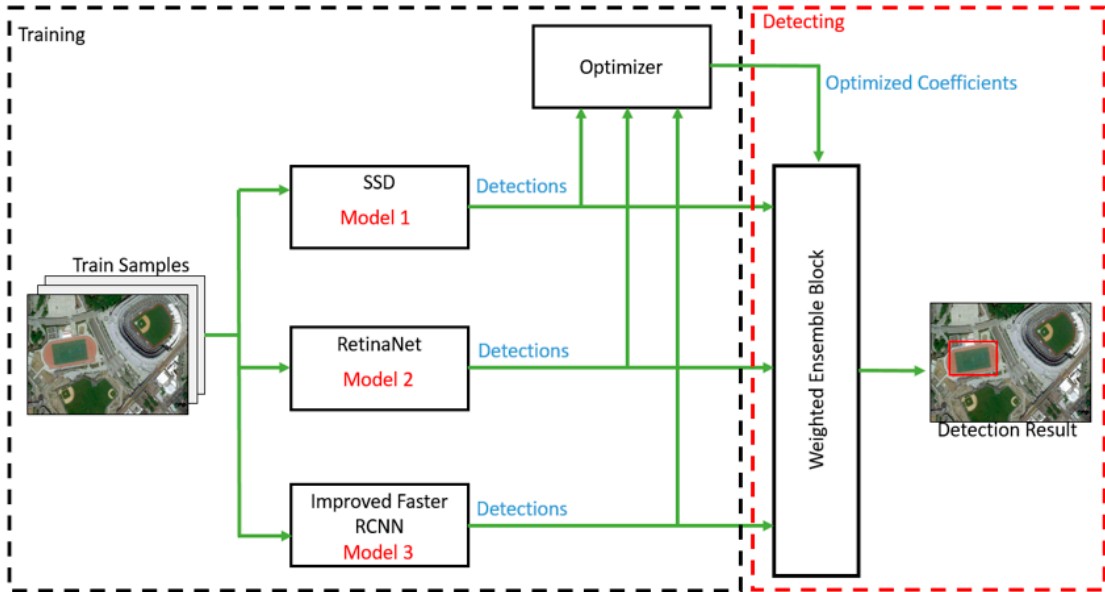

**Figure 4.** Block diagram of weighted ensemble object detection method with optimized coefficients.

### 3.1. Object Detection Models

In the proposed model, three different object detection models trained on the same dataset are used. These models are Single Shot Multibox Detector (SSD), RetinaNet, and Improved Faster R-CNN for remote sensing images. Improved Faster R-CNN is a two-stage object detection model, while SSD and RetinaNet are single-stage object detection models. These object detection models are independently trained. Details of the training process are described below:

- Twenty-four Epoch training was performed, and the batch size was 64;
- Learning rate = 0.0025 and momentum = 0.9. A gradual learning rate technique was applied by reducing the learning rate. This was done by dividing 10 by the 16th epoch and 22nd epoch;
- Batch Normalization was used for the SSD and RetinaNet models, and Weight Standardization was used for the Improved Faster R-CNN model;
- Transfer learning was utilized for the trained Resnet50 model.

### 3.2. Weighted Ensemble Block

This algorithm corresponds to the Weighted Ensemble Block in Figure 4. It takes three different object detection results as the input from this algorithm and produces a single object detection result. The object detection concept involves the task of determining the position and category of multiple objects in an image. An object detection model is a function that, when given an image, returns a list of detections ($D = [d_1, \dots, d_N]$), where each $d_i$ is given by a triple [$b_i$, $c_i$, $s_i$] that consists of a bounding box, $b_i$, the corresponding category, $c_i$, and the corresponding confidence score, $s_i$.

The input of our ensemble algorithm is a List of Detections (*LD*) for a given image using three different object detection models (*LD* = [$D_1$, $D_2$, $D_3$]). Each *LD* has different $b_i$, $c_i$, and $s_i$ values. The algorithm steps are as follows:

(1) The detections in *LD* are grouped according to the class values and the overlap of the bounding boxes. Taking the two bounding boxes as $b_1$ and $b_2$, the following *IoU* equation is used to find the overlapped region between them.

$$IoU(b_1, b_2) = \frac{area\ (\ b_1\ \cap\ b_2\ )}{area\ (b_1\ \cup\ b_2\ )} \tag{1}$$

(2) An IoU threshold value of 0.5 is frequently used in related object detection studies, including the studies mentioned in the introduction. Thus, bounding boxes with $IoU > 0.5$ are assigned to a new Output List (LO). The LO has $b_i$ and $S_i$ values for each detection. It is necessary to recalculate a common box (X1, Y1, X2, Y2) and confidence score (S) using the weight coefficients ($k_i$) for the multiple different boxes in the LO clusters. These calculations are made with the following equations:

$$S = \frac{k_1 * S_1 + k_2 * S_2 + k_3 * S_3}{k_1 + k_2 + k_3} \tag{2}$$

$$X1 = \frac{k_1 * X1_1 + k_2 * X1_2 + k_3 * X1_3}{k_1 + k_2 + k_3} \tag{3}$$

$$X2 = \frac{k_1 * X2_1 + k_2 * X2_2 + k_3 * X2_3}{k_1 + k_2 + k_3} \tag{4}$$

$$Y1 = \frac{k_1 * Y1_1 + k_2 * Y1_2 + k_3 * Y1_3}{k_1 + k_2 + k_3} \tag{5}$$

$$Y2 = \frac{k_1 * Y2_1 + k_2 * Y2_2 + k_3 * Y2_3}{k_1 + k_2 + k_3}. \tag{6}$$

When calculating the confidence score in Equation (2), the weight coefficients and multiplication results of all confidence scores are averaged. In the same way, the coordinates of the bounding box are calculated by multiplying the individual weight coefficients in each of them (Equations (3)–(6)).

(3) If least two of the detections in the LD overlap, the above operations are performed. The resulting ensemble detections are added to the Predictions List (PL). At this point, some detections may not overlap at all. Those with a confidence score of (c)> 0.5 are also assigned to the PL.

(4) The detections in the PL are considered the final predictions, and the weighted ensemble process ends.

### 3.3. Optimizing Coefficients

This is the process for calculating the optimal coefficients according to the detection results obtained from the three models we used. In this process, the coefficients that converge the X1, Y1, X2, and Y2 coordinate values of the three models closest to the X1, Y1, X2, and Y2 coordinate values of the labels in the test dataset are calculated via nonlinear programming [30]. In this process, optimization is achieved by considering all test data as a whole. The calculations are as follows:

$$\hat{Y}_i = \sum_{n=1}^{3} \frac{k_n * X_{in}}{k_{sum}} , \; X_{in} = (X1_n , \; Y1_n , \; X2_n , \; Y2_n) \tag{7}$$

$$k_{sum} = k_1 + k_2 + k_3 \tag{8}$$

$$Constraints : \sum_{n=1}^{3} k_n = 1 , \; k_n \geq 0 \tag{9}$$

$$Z_{min} = \sum_{i=1}^{K} \left( Y_i - \hat{Y}_i \right)^2. \tag{10}$$

In Equation (8), $\hat{Y}_i$ represents the sum of the estimates, $k_n$ denotes the coefficients, and $X_{in}$ refers to the coordinate values (X1, Y1, X2, Y2) of the $i$th prediction of the $n$th model. Equation (9) is the formula for the sum of coefficients, and Equation (10) is the optimization objective function that minimizes the sum of the squares of the difference between the real coordinate values and the estimated coordinate values. The constraints of this optimization objective function are given in Equation (9). For the optimization objective function to find the minimum value, the Mean Squared Deviation

(MSD) [31] technique is used. The values of $k_1$, $k_2$, $k_3$ that provide the $Z_{min}$ value are considered the optimal coefficients.

## 4. Results and Discussion

An experimental setup environment was created to evaluate the performance of the proposed model on the NWU-VHR10 and RSOD datasets. Three object detection models used as the input for the ensemble model and ensemble object detection results under different combinations of coefficients were compared. Then, a comparison was carried out with other related studies using the NWPU-VRH10 and RSOD datasets to evaluate how successful the proposed model was.

### 4.1. Environment and Evaluation Criteria

For the physical environment, the PC used for the experiments features an Intel®Core ™ i5 2.4 GHz CPU, with 6 GB RAM, a Geforce GTX 1080 graphics card, and the Ubuntu 16.04 LTS operating system. The proposed model was written in the python programming language using the pytorch deep learning library. To evaluate the performance of the proposed model, the precision–recall curve (PRC) and mean average precision (*m*AP) [32] criteria, which are used to measure the success of many object detection models [33–39] in this area, are used.

- *Precision–Recall Curve (PRC):* Precision is used to indicate the accuracy of the detection rate of true positive values, and recall is used to indicate the ratio of true positive values detected correctly [40]. The number of true positives is TP, the number of false positives is FP, and the number of false negatives is FN. The precision–recall curve value is calculated with the following formulas:

$$Precision = \frac{TP}{(TP + FP)} \tag{11}$$

$$Recall = \frac{TP}{(TP + FN)}) \tag{12}$$

where the TP value used in Equations (11) and (12) refers to the situations where the area between the box overlaps, and the Ground Truth (GT) box is greater than 0.5 as a result of the detection. The opposite case is considered to be the FP. Further, if multiple detections coincide with the same GT box, only one is considered the TP, and the others are FP.

- *Mean Average Precision (mAP):* This can be explained as the average of the precision values of all classes in the area under the PRC. Therefore, the performance of an object detection model is measured by the height of the *m*AP value. A higher *m*AP value indicates better performance.

### 4.2. Dataset Preparation Phase

In the experiments, the NWPU-VHR10 dataset [41] and RSOD dataset [42] were used. These datasets are described below.

*(1) NWPU-VHR10 dataset:* As its name suggests, this dataset has 10 classes (aircraft, ship, storage tank, baseball diamond, tennis court, basketball court, highway court, port, bridge, and vehicle). Moreover, this dataset contains 650 positive images and 150 negative images (total 800 images), whose spatial resolution ranges from 0.5 to 2 m.

In the experiments, positive images in the dataset were used in two different ways (60% training—390 images, 20% verification—130 images, 20% testing—130 images, and 50% training—325 images, 20% verification—130 images, 30% testing—195 images). The set of 650 images containing objects consists of 565 images from Google Earth with a spatial resolution ranging from 0.2 m to 2 m and 85 sharpened images with a spatial resolution of 0.08 m from the Vaihingen dataset [43].

*(2) RSOD dataset:* The RSOD dataset contains 976 images downloaded from Google Earth and Tianditu [44]; the spatial resolutions of these images range from 0.3 to 3 m. This dataset consists of 6950

total object instances covered by four object classes, including 1586 oil tanks, 4993 airplanes, 180 overpasses, and 191 playgrounds. In the experiments, images in this dataset were used in two different ways, as in the NWU-VHR10 dataset (60% training—586 images, 20% verification—195 images, 20% testing—195 images, and 50% training—488 images, 20% verification—195 images, 30% testing—293 images).

The training objects in the NWU-VHR10 and RSOD datasets are few in number, which directly affects the success of the model. Data augmentation [45] was used to eliminate this disadvantage. The operations applied in the data augmentation process to the images in the datasets are presented sequentially in Figure 5.

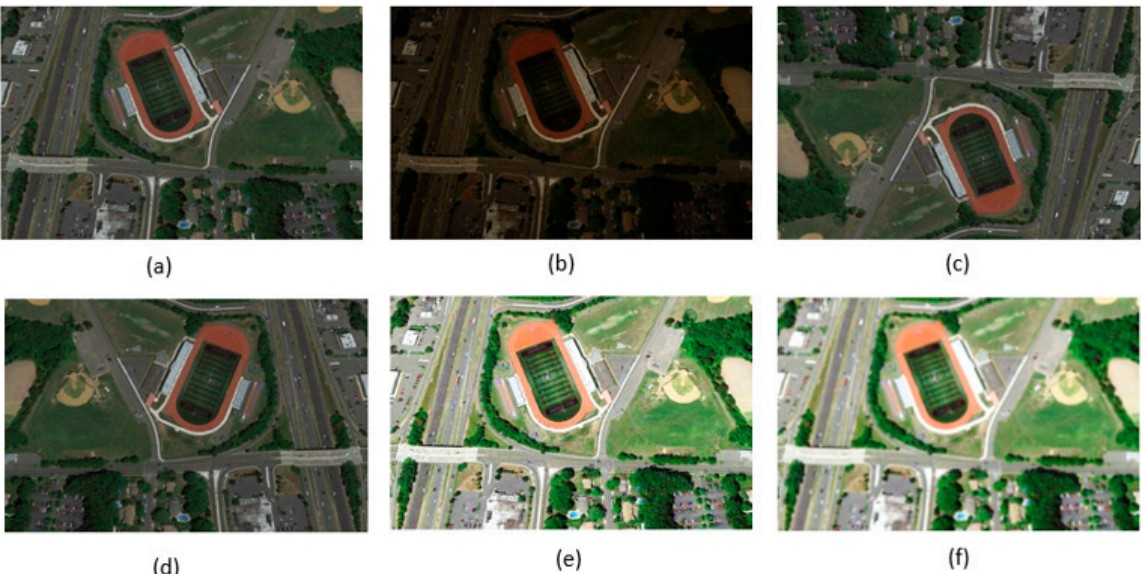

**Figure 5.** Data augmentation examples: (**a**) original picture; (**b**) blur; (**c**) vertical rotation; (**d**) horizontal rotation; (**e**) gamma conversion; (**f**) random image brightness.

### 4.3. Results with Different Coefficient Combinations and Optimal Coefficient Values

For the NWPU-VHR10 and RSOD datasets, the three object detection models used as the input for the ensemble model and the ensemble object detection results made with the different combinations and optimal values of the coefficients were compared. To better evaluate the performance of the proposed model, the comparison process was done according to two different train–test ratios frequently used in the literature. A comparison of the results is given in Tables 1 and 2.

**Table 1.** Comparison of the results of three different object detection models and different coefficient combinations for the Northwestern Polytechnical University Very High Resolution 10 (NWPU-VHR10) dataset.

| Model | Coefficients | *m*AP (70% Training—30% Testing) | *m*AP (80% Training—20% Testing) |
|---|---|---|---|
| SSD | - | 0.832 | 0.790 |
| Imp. Faster R-CNN | - | 0.916 | 0.889 |
| RetinaNet | - | 0.920 | 0.881 |
| Experiment 1 | $k_1 = k_2 = k_3 = 1$ | 0.931 | 0.913 |
| Experiment 2 | $k_1 = 2, k_2 = k_3 = 1$ | 0.924 | 0.909 |
| Experiment 3 | $k_1 = k_3 = 1, k_2 = 2$ | 0.933 | 0.901 |
| Experiment 4 | $k_1 = k_2 = 1, k_3 = 2$ | 0.930 | 0.905 |
| Experiment 5 | $k_1 = k_2 = 2, k_3 = 1$ | 0.929 | 0.908 |
| Experiment 6 | $k_1 = k_3 = 2, k_2 = 1$ | 0.932 | 0.907 |
| Experiment 7 | $k_2 = k_3 = 2, k_1 = 1$ | 0.927 | 0.906 |
| Experiments 8 * | $k_1 = 0.0226, k_2 = 0.6527, k_3 = 0.3246$ | 0.942 | - |
| Experiments 9 ** | $k_1 = 0, k_2 = 0.7982, k_3 = 0.2017$ | - | 0.921 |

* Optimal Coefficients for 70% Training—30% Testing ** Optimal Coefficients for 80% Training—20% Testing.

**Table 2.** Comparison of the results of three different object detection models and different coefficient combinations for the Remote Sensing Object Detection (RSOD) dataset.

| Model | Coefficients | *m*AP (70% Training—30% Test) | *m*AP (80% Training—20% Testing) |
|---|---|---|---|
| SSD | - | 0.869 | 0.860 |
| Imp. Faster R-CNN | - | 0.895 | 0.884 |
| RetinaNet | - | 0.907 | 0.890 |
| Experiment 1 | $k_1 = k_2 = k_3 = 1$ | 0.922 | 0.901 |
| Experiment 2 | $k_1 = 2, k_2 = k_3 = 1$ | 0.917 | 0.902 |
| Experiment 3 | $k_1 = k_3 = 1, k_2 = 2$ | 0.923 | 0.900 |
| Experiment 4 | $k_1 = k_2 = 1, k_3 = 2$ | 0.918 | 0.901 |
| Experiment 5 | $k_1 = k_2 = 2, k_3 = 1$ | 0.921 | 0.904 |
| Experiment 6 | $k_1 = k_3 = 2, k_2 = 1$ | 0.914 | 0.902 |
| Experiment 7 | $k_2 = k_3 = 2, k_1 = 1$ | 0.917 | 0.899 |
| Experiments 8 * | $k_1 = 0.0553, k_2 = 0.3004, k_3 = 0.6442$ | 0.949 | - |
| Experiments 9 ** | $k_1 = 0.0196, k_2 = 0.1602, k_3 = 0.8200$ | - | 0.921 |

* Optimal Coefficients for 70% Training—30% Testing ** Optimal Coefficients for 80% Training—20% Testing.

In Tables 1 and 2, k1, k2, and k3 are the coefficient values for SSD, RetinaNet, and the Improved Faster R-CNN models, respectively.

The most successful results were obtained from the experiments using the optimal coefficients. Further, all ensemble coefficient combinations achieved high success in the single object detection models.

*4.4. Comparing Proposed Model with Other Studies*

In this section, we compare the proposed model to other studies that use both the NWPU-VHR10 dataset and the RSOD dataset.

(1) Comparision of the NWPU-VHR10 dataset: A comparison with other studies that used the NWPU-VRH10 dataset in this field was carried out to evaluate how successful the proposed model was. The comparison results are given in Table 3. The most successful results per class are marked with underlined bold font in the table.

(2) Comparision of the RSOD dataset: A comparison was made with other studies in this field that used the RSOD dataset to assess how successful our proposed model was, like with the NWPU-VHR10 dataset. Existing studies using the RSOD dataset are less common than those using the NWPU-VHR 10 dataset. The comparison results are given in Table 4. The most successful results per class are marked with underlined bold font in the table.

**Table 3.** Success of diverse models on the NWPU-VHR10 dataset.

| Class | RICNN [2] | CACNN [6] | DFCCNN [7] | FMSSD [8] | COPD [10] | RICAO [11] | DODN [12] | Improved Faster R-CNN [13] | Our Study |
|---|---|---|---|---|---|---|---|---|---|
| Plane | 0.8835 | 0.9991 | 0.9085 | 0.9970 | 0.6225 | 0.9970 | 0.9392 | 0.9997 | **1.0000** |
| Ship | 0.7734 | 0.9055 | 0.9011 | 0.8990 | 0.6937 | 0.9080 | 0.9297 | 0.9458 | **0.9780** |
| Storage Tank | 0.8527 | 0.9001 | 0.8768 | 0.9030 | 0.6452 | 0.9061 | **0.9925** | 0.3670 | 0.6982 |
| Baseball Diamond | 0.8812 | 0.9965 | 0.9882 | 0.9820 | 0.8213 | 0.9291 | 0.9633 | **0.9691** | 0.9680 |
| Tennis Court | 0.4083 | 0.9016 | 0.8950 | 0.8600 | 0.3413 | 0.9029 | **0.9612** | 0.9306 | 0.9310 |
| Basketball Court | 0.5845 | 0.9091 | 0.9078 | 0.9680 | 0.3525 | 0.8013 | 0.7097 | 0.9295 | **0.9681** |
| Ground Track Field | 0.8673 | 0.9091 | 0.9062 | 0.9960 | 0.8421 | 0.9081 | **1.0000** | 0.9989 | 0.9970 |
| Harbor | 0.6860 | 0.8897 | 0.8872 | 0.7560 | 0.5631 | 0.8029 | 0.9688 | 0.9381 | **1.0000** |
| Bridge | 0.6151 | 0.7962 | 0.9034 | 0.8010 | 0.1643 | 0.6853 | 0.8115 | 0.6871 | **0.9030** |
| Vehicle | 0.7110 | 0.8900 | 0.8773 | 0.8820 | 0.4428 | 0.8714 | 0.8009 | 0.9328 | **0.9340** |
| *m*AP | 0.7263 | 0.9097 | 0.9052 | 0.9040 | 0.5489 | 0.8712 | 0.9077 | 0.8710 | **0.9420** |

Using our weighted ensemble object detection model, we performed object detection for all classes in the test dataset. The object detection results for the test images of the proposed model are provided in Appendix A. In Appendix A, true positive detections are marked in green, false-positive detections are marked in red, and false negative detections are marked with a blue box.

**Table 4.** Success of diverse models on the RSOD dataset.

| Class | SigNMS [5] | A$^2$RMNet [14] | DConvNet [15] | RFN [16] | Our Study |
|---|---|---|---|---|---|
| Aircraft | 0.806 | 0.942 | 0.718 | 0.791 | **0.947** |
| Oil Tank | 0.906 | **0.964** | 0.903 | 0.905 | 0.936 |
| Overpass | 0.874 | 0.838 | 0.895 | **1.000** | 0.912 |
| Playground | 0.991 | 0.997 | 0.998 | 0.997 | **1.000** |
| *m*AP | 0.894 | 0.935 | 0.879 | 0.923 | **0.949** |

### 4.5. Discussion

In this study, three parameters that can affect the object detection performance of the model relative to the dataset were considered. These parameters are the number of images containing the object in the dataset, the number of objects in each image, and the size of the object in the image. As noted in other studies [2,5,8,10,16] (using the comparisons in Tables 3 and 4), the numbers of positive image sets from the NWPU-VHR10 dataset (650 images) and the images of the RSOD dataset (976 images) were too few for deep learning object detection models. We aimed to solve this problem with data augmentation. However, data augmentation alone was not enough to eliminate class imbalance. To eliminate the imbalance between classes, we combined the strengths of these models by using coefficients rather than designing a separate loss function for these three object detection models. Each model was trained independently using its own loss function. Then, the optimal coefficients were calculated using the MSD technique. In this way, the proposed model detects objects using the optimal coefficient values. In our experiments, we observed the effects of these coefficient changes on success.

As shown in Tables 1 and 2, the optimal ensemble coefficients achieved greater success than the standalone object detection models that we used as input for the proposed model. In addition, other ensemble coefficient combinations achieved excellent success using single object detection models. In the conducted experiments, the proposed model increased the performance of classification for the NWPU-VHR10 and RSOD datasets by 11% to 16.5% for SSD, 0.78% to 6% for RetinaNet, and 0.88% to 4.54% for Improved Faster R-CNN. As shown by the experimental results, the highest improvement in classification performance was achieved with the SSD model because our model eliminated the disadvantages of the SSD model by applying the RetinaNet and Improved Faster R-CNN models, which are more successful than the SSD model.

As mentioned earlier, the small number of images found in the NWPU-VHR10 and RSOD datasets, which are frequently used in remote sensing literature, is too low for deep learning models. Thus, while greater success would be expected by training the model with more data under normal conditions, in our experiments with 80% training data, lower values were obtained than in experiments with 70% training data. This situation can be explained in two ways:

- Overfitting: The proposed model experienced some overfitting in experiments conducted at a rate of 80% training, despite the use of data augmentation;
- The Size of the Test Data: The object detection success of the model could be better measured by testing with more data. Accordingly, better results were obtained in tests with 30% testing data.

The results given in Table 3 reveal that the proposed model provides better results than those in other studies. Although our method ensures high performance, we observed a discrepancy in the detection accuracy for the same category of object (e.g., storage tank) as in the DODN [12] and Improved Faster R-CNN [13] models. This is mainly because of the imbalance among the classes in the NWPU-VHR10 database. The object detection rate of the classes (e.g., planes) with a large training dataset is high, while the object detection success of the classes (e.g., storage tank) with a small training dataset is quite low. Although data augmentation improved the training performance, its contribution remains limited. Table 4 shows that the proposed model did not experience this problem with the RSOD dataset. This is because the amount of image data used to train deep learning models

directly affects the object detection success of the model (the RSOD dataset contains 50% more images (325 images) than the NWPU-VRH10 dataset).

Despite the inadequacies of this dataset, the proposed model achieved high success in detecting objects of different orientations and sizes in the test images. Moreover, our model resulted in higher performance results than those in other studies using the same dataset for object detection. The proposed model achieved a higher *m*AP than other studies in six of the ten object classes in the NWPU-VHR10 dataset, while in the RSOD dataset, our model achieved higher *m*AP success in two of the four object classes. We believe that the success of the proposed model is the result of the following:

- Our model combines the powerful features of three different models (SSD, RetinaNet, and Improved Faster R-CNN), which have proven themselves successful for object detection;
- We calculated the optimal coefficients of the object detection models using the MSD technique;
- Each object detection model is subjected to an independent training process, so there is no overfitting problem.

## 5. Conclusions

In this study, an object detection model that performs the optimized coefficient-weighted ensemble operations for the outputs of three different object detection models trained on the same dataset was proposed. The proposed model was designed to provide an output with an optimized coefficient-weighted ensemble by taking two or more object detection methods as its input. In addition to providing an ensemble output by multiplying the most successful determinations of the object detection models used by the input with the coefficients determined through optimization using the MSD technique, the results obtained under various coefficient changes were more successful than the results with the object detection models used as input. The object detection performance of our proposed model was assessed using the NWPU-VHR10 and RSOD datasets. The highlights of our model include combining the strengths of SSD, RetinaNet, and Improved Faster R-CNN models as inputs and training these models separately to prevent overfitting. Therefore, the results of the experiments reveal that our model can obtain better results than current models that use the same datasets. In summary, no matter how different the models used as inputs and the hyperparameters of these models are, our ensemble model, which can use the strengths of the input models most effectively through optimal coefficients, achieves better results than the models we used as inputs.

## 6. Future Work

In our future work, studies will be continued on the following topics:

- Different models and their experimental results will be analyzed by multiplying these models with different coefficients to develop a dataset to measure the success of the coefficients used for the model. Then, a simple neural network will be trained with this dataset. Using this training, we will attempt to reveal the predicted success of the network for the purpose of determining the most successful weighted ensemble method with machine learning. We will also test how successful the final results of the machine learning are through experiments using the proposed model;
- To increase the object detection success of our model, a future study will be carried out focusing on reducing the effects of the imbalance between classes by designing the loss functions of the object detection methods (SSD, RetinaNet, and the Improved Faster R-CNN) as a single loss function.

**Author Contributions:** Conceptualization, Atakan Körez, Necaattin Barışçı, Aydın Çetin, and Uçman Ergün; data curation, Atakan Körez, Necaattin Barışçı, Aydın Çetin, and Uçman Ergün; formal analysis, Atakan Körez, Necaattin Barışçı, Aydın Çetin, and Uçman Ergün; investigation, Atakan Körez, Necaattin Barışçı, Aydın Çetin, and Uçman Ergün; methodology, Atakan Körez, Necaattin Barışçı, Aydın Çetin, and Uçman Ergün; project administration, Necaattin Barışçı and Aydın Çetin; resources, Atakan Körez, Necaattin Barışçı, Aydın Çetin, and Uçman Ergün; software, Atakan Körez; supervision, Necaattin Barışçı; validation, Atakan Körez, Necaattin Barışçı, Aydın Çetin, and Uçman Ergün; visualization, Atakan Körez and Uçman Ergün; writing—original

draft, Atakan Körez, and Necaattin Barışçı; writing—review and editing, Atakan Körez, Necaattin Barışçı, and Aydın Çetin. All authors have read and agreed to the published version of the manuscript.

**Funding:** There was no external funding.

**Acknowledgments:** We would like to thank TÜBİTAK ULAKBİM High Performance and Grid Calculation Center for the high calculation servers we have tested the model created within the scope of the study. The authors thank all the reviewers for their comments, which improved the quality of the paper.

**Conflicts of Interest:** The authors declare no conflict of interest.

## Appendix A. Object Detection Results of the Weighted Ensemble Object Detection Model

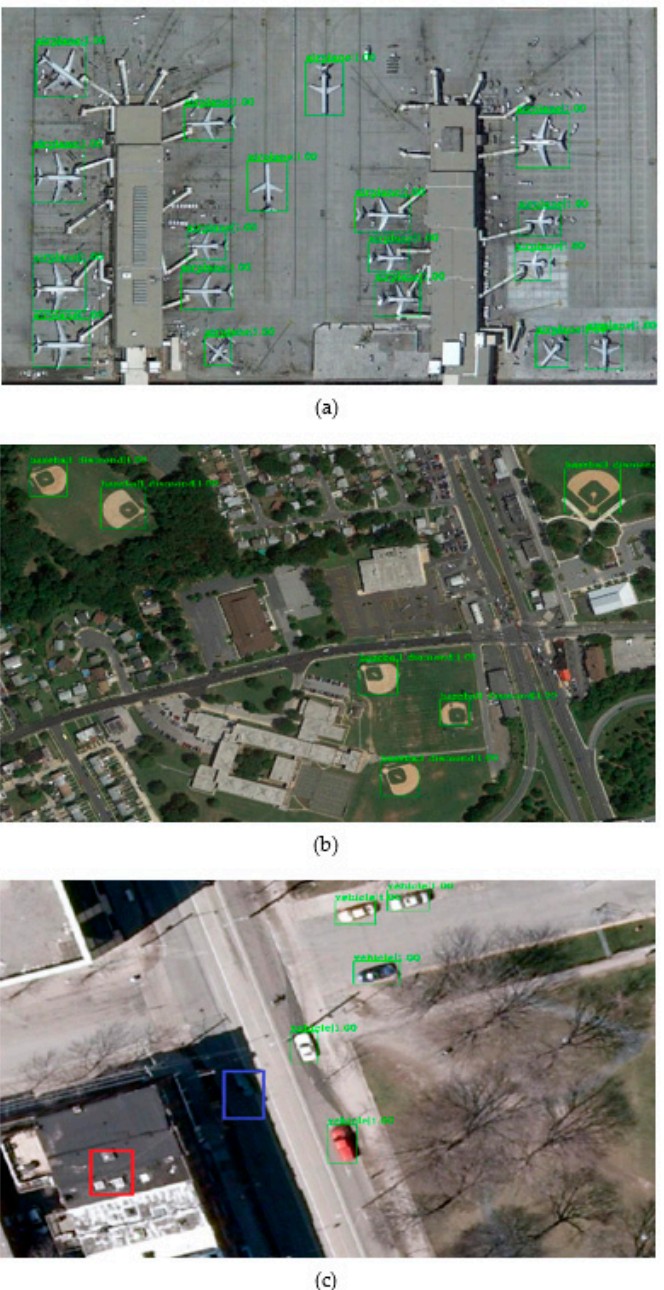

**Figure A1.** Object detection examples on the NWPU-VHR10 dataset: (**a**) plane; (**b**) baseball diamond; (**c**) vehicle.

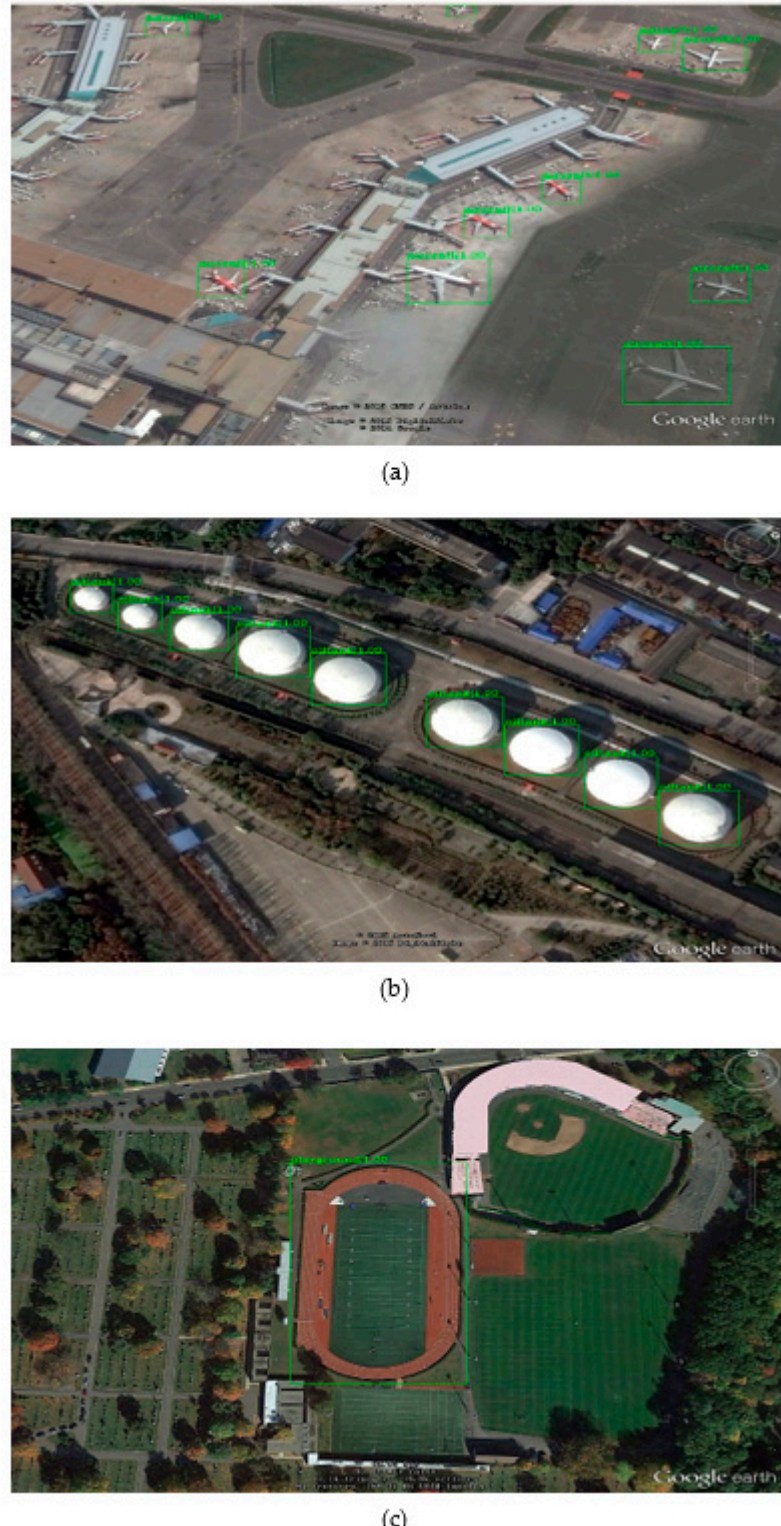

**Figure A2.** Object detection examples on an RSOD dataset: (**a**) aircraft; (**b**) oil tank; (**c**) playground.

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
