# Peer review of "Weighted Ensemble Object Detection with Optimized Coefficients for Remote Sensing Images"

_ijgi, doi:10.3390/ijgi9060370_

Round 1

Reviewer 1 Report

Thank you for the continual improvements to the manuscript but i feel it is still lacking evidence that the ensemble object detector is provably better than the retinanet model.  

  • It is unclear how train/test/val where used in each of the training cases
  • How were the 3 base models trained and were their hyperparameters optimized?
  • The new experiments 8 and 9 use additional data, ie, the test data where the k1,k2,k3 were optimized over.  These results thus cannot be compared to the SSD, FRCNN, or RetinaNet which did not have "hyperparameter tuning" performed on them.  In Table 2 the only improvement came from the non-integer k1,k2,k3 values so i cant make any judgements from this experiment. 
  • In Table 1/2 why do the MAP scores fall with more training data is used?

This ensemble technique while interesting lacks evidence that it is better than a properly trained initial model.  I think what you need to do is show that the SSD, FRCNN, and RetinaNet are first optimally trained for the datasets by optimizing the hyperparameters with the test dataset.  Then show that the "optimal" models with the k1,k2,k3 optimized over the same test set perform statistically better on the val set. 

Author Response

Dear Reviewer,

Thank you for comments concerning our manuscript entitled “Weighted Ensemble Object Detection with Optimized Coefficients for Remote Sensing Images”. Those comments are all valuable and very helpful for revising and improving our manuscript, as well as the important guiding significance to our studies. We have studied comments carefully and have done all corrections. Reponses to comments have been added as attachment.

Reviewer 2 Report

Dear Authors,

Thank you for providing us with the response and corrections. At the moment, I do not have any further feedback. I hope this manuscript can contribute valuable inputs to readers. Thank you very much for your work.

Best regards.

Author Response

Thank you for your comments that improve the quality of our article. Best regards.

Reviewer 3 Report

edits done. The article is good to be printed as it is

Author Response

Thank you for your comments that improve the quality of our article. Best regards.

This manuscript is a resubmission of an earlier submission. The following is a list of the peer review reports and author responses from that submission.

Round 1

Reviewer 1 Report

very nice article, short but clear and concise.

please expand the abbreviations the first time they appear on the article:

line 15/  NMPU-VHR10 

line 17 mAP

line 33 CNN(RICNN)

line 36 Sig-NMS

I think those are all, the rest are already expanded but please check.

Author Response

Dear Reviewer,

Thank you for comments concerning our manuscript entitled “Weighted Ensemble Object Detection for Remote Sensing Images”. Those comments are all valuable and very helpful for revising and improving our manuscript, as well as the important guiding significance to our studies. We have studied comments carefully and have done all corrections. Reponses to comments have been added as attachment.

Reviewer 2 Report

Thank you for the manuscript, I guess it is good for new knowledge. However, there are many questions to make sure, thus, it will make the manuscript more understandable for the readers.

  1. Please explain what is NWPU stands for (in line 15)? I tried to find out in http://www.escience.cn/people/gongcheng/NWPU-VHR-10.html as suggested in Reference number 23 but It was difficult to find. Please explain what is CNN (line 33), NMS (line 36), GPU (line 51) stand for?
  2. It is difficult to understand this sentence on line 57:

“In the scope of the idea that a flexible and training-independent ensemble operation would be more practical and effective; a weighted ensemble object model is proposed, using 3 different object detection models, independently trained on the same data set.”

There is a gap idea between the previous sentence and this sentence. Why would a word “weighted” is emerged and proposed suddenly? Please add some sentence(s) and also supported reference(s) to fill the gap.

I guess the details on “...using 3 different object detection models, independently trained on the same data set.” I guess is not part of the introduction, but instead, part of method. Unless you provide more context related to this on the previous paragraph.

In the introduction, I guess it is still lack of idea supporting the research.

  1. line 79, why would you use the three models? Why they are so important comparing to other models? Thus, the readers can absorb the insight. Please add the reasons to the manuscript. The explanation regarding these models are not enough for readers. Please give deeper understanding to each of the models
  2. Figure 1 and line 90, It is not clear which of the three models use each of the three objects. Did you use same dataset or not for the three objects for each of the three models? Why would the picture position of ‘Detection Result’ on the figure one is different with the train sample? Line 94, what is D stands for, detections? Please make it clear. Please explain what is bounding box to make it easier for the readers of this manuscript.
  3. Line 104, please put reference and reason why did you put 0.5 cut off. Do many other studies use this cut off too? Please provide more information and also reference(s).
  4. Line 105, what is common box, is it position or latitude longitude? Please explain it. What is confidence score? Does Confidence score has cut off too? Please put reference.
  5. Line 123, It is not consistent with previous information regarding “confidence score of (c)> 0.5” It makes reader confused reading this information.
  6. Line 122, why using word discarded?
  7. Please explain more about validation including using training data and ground truth validation process to identify the result quality. Did you use confusion matrix to get accuracy (Producer accuracy based on the reference data, and User accuracy based on the class data, as well as Overall accuracy) and Kappa coefficient to identify the accuracy cut off? If you don’t use confusion matrix, but instead, PRC and mAP, please explain why and provide reference(s) as well.
  8. Line 133, to make reader understand the interpretation, please add the precision criteria and also its reference(s).
  9. Line 153, Please explain what is positive and negative images? And why and how you used these?
  10. Line 155, please add numbers of data for training, validation (reference data), and test, not only percentage.
  11. Line 157, “Data Augmentation is used to eliminate this disadvantage” please add reference to this information.
  12. Line 171-173, “the coefficients of one model are two times the coefficients of the other two models and the coefficients of two models are equal to two times the coefficient of the third model.” I am not sure I can understand this sentence. Which model refers to which model on the Table 1. Please make it clearer to make readers happy to understand.
  13. On table 2, it is mAP for all in the Class column? If yes, it is ok
  14. On figure 3:

This picture is taken from what data imagery? Because to my understanding, its resolution seems clearer (or less than 0.5 m per pixel) looking at, for example, the vehicles. For example, pan-sharpened Worldview 2 Imagery resolution is 0.5 m (0.46 m for the exact) resolution, and when we zoom a vehicle, we still can see pixels constructing it, but not as clearer as above picture. You wrote on line 154 that the images ranging from 0.5 to 2 meters, therefore, maximum higher resolution according this is 0.5 m, but the picture above seem does not, to my opinion. Please give more information with regard to this picture to make reader more understand.

  1. I do not see where is discussion on the Discussion part. It is very important part. This is the part where you compare your study to other previous studies, where you provide logical reasons based on references why your study has results like this and that. For example, on line 195-196,” Our model is very successful in detecting many objects of different orientation and sizes in test images.” How can you be sure that your model is very successful if you do not provide comparison and logical reasons supported by previous studies.
  2. Conclusions part is also viewed based on discussions. If no discussion, how can we assure the conclusions.

Thank you very much for the opportunity to review the manuscript.

Best Regards.

Author Response

(The authors gave the same response as above.)

Reviewer 3 Report

While the methodology is interesting the synthesis of three different object models does not produce the scientific originalism necessary for a journal publication.  There is no explanation as to why these models would produce different results or differ on basis of class or scale.  The advantage of the combined model is small in comparison to the RCNN. 

I also find flaw in the statements around using data augmentation to address class imbalance.  This method will not adequately address the problem and the authors should look to weighting the loss function on a class specific basis.  

Author Response

(The authors gave the same response as above.)

Round 2

Reviewer 2 Report

Dear Authors,

Thank you very much for providing the revisions. To complete, the Discussion part still needs references. I do not see sentences that cite some or many references on this part. To my opinion, it is crucial to assure and or compare Authors’ results with other previous researches. I, as a reader, and I believe other readers, would like to see that to enrich our points of view, to stimulate discussion and insight in our mind. Therefore, I would like to encourage Authors to put also what other researchers say related to the content of discussion part.

Thank you very much for your work.

Best Regards.

Reviewer 3 Report

While the paper is well written I don't see the novelty in weighted combination of several object detection models which have all been applied to remote sensing data before. 

  • There is no evaluation to the brittleness of the k- coefficients used to combine the models and the improvement is quite small FRCNN = .889, k1=k2=k3 .913. Can you get similar results by tuning some of the model parameters like the box scales and number of proposals?
  • It would have been interesting to treat this as an optimization or multi-objective optimization problem and find the optimal set of coefficients. 
  • The authors dont go and test these finds on another dataset - are these results consistent or do you need to learn another set of ks for each collect?  
  • Similar work has been done in 2018 and the method to combine the models is more complicated: https://arxiv.org/pdf/1803.06554.pdf